# Review Article: A comprehensive review of datasets and methodologies employed to produce thunderstorm climatologies

Leah Hayward[1], Dr Malcolm Whitworth[1], Dr Nick Pepin[1], Professor Steve Dorling[2]

[1] School of the Environment, Geography and Geosciences, University of Portsmouth, Burnaby Building, Burnaby Road, Portsmouth, PO1 3QL, United Kingdom

[2] School of Environmental Sciences, University of East Anglia, Norwich Research Park, Norwich, NR4 7TJ, United Kingdom.

*Correspondence to:* Leah Hayward (leah.hayward@port.ac.uk)

**Abstract.** Thunderstorm and lightning climatological research is conducted with a view to increasing knowledge about the distribution of thunderstorm related hazards and to gain an understanding of environmental factors increasing or decreasing their frequency. There are three main methodologies used in the construction of thunderstorm climatologies; thunderstorm frequency, thunderstorm tracking or lightning flash density. These approaches utilise a wide variety of underpinning datasets and employ many different methods ranging from correlations with potential influencing factors and mapping the distribution of thunderstorm day frequencies, to tracking individual thunderstorm cell movements. Meanwhile, lightning flash density climatologies are produced using lightning data alone and these studies therefore follow a more standardised format. Whilst lightning flash density climatologies are primarily concerned with the occurrence of cloud to ground lightning, the occurrence of any form of lightning confirms the presence of a thunderstorm and can therefore be used in the compilation of a thunderstorm climatology. Regardless of approach, the choice of analysis method is heavily influenced by the coverage and quality (detection efficiency and location accuracy) of available datasets as well as by the controlling factors which are under investigation. The issues investigated must also reflect the needs of the end use application to ensure that the results can be used effectively to reduce exposure to hazard, improve forecasting or enhance climatological understanding.

## 1. Introduction

Thunderstorms have the potential to produce hazardous weather. All thunderstorms produce lightning, whilst the presence of other weather hazards such as wind, hail, heavy rain and snow can vary with geographic, climatic and synoptic conditions. The intensity of these hazards may vary by region and time of the year and, indeed, from storm to storm. This hazardous weather can cause flooding; damage to property, infrastructure and crops; disruption to transport and outdoor maintenance; injury and a threat to life (Elsom et al., 2018; Piper et al., 2016). One example was the death of a hiker on a ridge in Glencoe, Scotland in June 2019 (Woman dies after being struck by lightning in Scottish Highlands , 2019). The July 2019 Latitude Festival in England was halted for an hour for safety reasons due to local lightning risk ( Latitude Festival, 2019) and in that same month 7 deaths, 140 injuries and severe damages were caused by a thunderstorm in Greece with high winds, hail and intense rainfall overturning cars, felling trees, causing flooding and damaging houses and roofs (Freak storm kills seven in Greece, 2019).

Figure 1 is a Venn diagram of weather hazards in a convective cell. This shows that all thunderstorm convective cells must produce lightning to distinguish them from an ordinary convective cell (Doe, 2016). Where

precipitation or wind hazards occur without lightning, they are the result of non-electrical convective activity and beyond the scope of this review.

Thunderstorm climatology research usually falls into one of three categories; thunderstorm frequency, thunderstorm tracking and lightning flash density (lightning strikes per square kilometre per year). Studies may sometimes utilise more than one approach and thus boundaries between the three can be blurred. Whilst thunderstorm frequency and tracking are concerned with the thunderstorm as a whole and all the hazards therein, lightning flash density is usually concerned exclusively with cloud to ground lightning hazards. Intra-cloud and cloud to cloud lightning strikes are not included because the focus of such work is on the risk to human life, property and industry. Lightning flash density and lightning frequency are however a form of thunderstorm climatology, because lightning is the only product of a thunderstorm which is unique to its diagnosis.

Producing and communicating the results of thunderstorm climatologies increases public and expert understanding of thunderstorm hazards and how to best reduce associated risks (Brooks et al., 2018). They provide important information for those who may be most exposed to thunderstorm hazards such as outdoor workers and those pursuing outdoor recreation as well as industries which may be vulnerable to disruption such as the power sector, construction and farming (Elsom and Webb, 2017). Preparedness may take different forms, from planning the most appropriate time of year to conduct outdoor maintenance or the most appropriate time of the day to start a hike, to local authorities ensuring that drains and other defences are working efficiently prior to the most thunderstorm active times of year.

Accurately diagnosing the weather hazards that are the direct result of thunderstorms can be a challenge, because other than lightning, some precipitation and wind hazards can also be present without a thunderstorm. To ensure the correct diagnosis of thundery convection and the accurate assessment of the spatial and temporal distribution of thunderstorms, climatologists utilise a variety of datasets and methods. Choosing the most appropriate analysis approach and dataset is key to obtaining results that a) best reflect the distribution of the hazard concerned and b) are useful to the intended end user.

The purpose of the paper is to conduct a systematic and comprehensive review of the datasets and methodologies applied to create thunderstorm climatologies. This review aims to assist those at the design stage of their research and those new to the subject area to become familiar with the strengths and weaknesses of the available data types, to consider which climatological approach best fits their research goal and to identify potential alternative approaches which may not have previously been considered. Whilst there are existing reviews in this subject area available (Betz et al., 2009; Cummins and Murphy, 2009; Ellis and Miller, 2016; Nag et al., 2015), these tend to focus either on analysis of a particular dataset, data type or methodology. This paper, in contrast, fills a gap in the literature by providing an overview of the whole subject area to help the reader to subsequently move on to more specific and detailed examples. Lastly, recommendations for research areas which require development are made.

To fulfil the above purposes, we first review the dataset types in section 2, before then moving on to evaluating how different dataset types have been applied in compiling thunderstorm frequency climatologies (section 3) and thunderstorm tracking (section 4). Section 5 reviews the methods used to produce lightning flash density climatologies, using one dataset type: lightning remote sensing data. This section also includes a review on how

lightning flash density results have correlated with potential drivers of thunderstorm formation, such as topography, which thereby introduces further methods and datasets. Recommendations for study design are contained in Section 6 and future research areas outlined in Section 7.

## 2. Data

Thunderstorm climatologies have traditionally been compiled and analysed using records kept by spotter networks which report thunder heard and lightning seen in different locations (Enno, 2015). Technology has progressed to include radar, satellite sensing and lightning location networks. As a result, research has developed to include information such as; cell movement (Lock and Houston, 2015); hazard intensity (Ellis and Miller, 2016); and spatial and temporal extent (Galanaki et al., 2018). Tables 1 to 4 provide a summary of strengths and weaknesses of the main dataset types discussed below. Figure 2 provides a checklist of issues to consider when choosing an appropriate dataset. In the following discussion, for each of the three main approaches, we consider the use of different dataset types including manual reports, radar and satellite approaches and model reanalyses.

### 2.1. Manual records: spotter networks and archives

Spotter networks can range from professional observations, such as weather records made at airports (Pinto, 2015), to crowdsourcing reports from enthusiasts, experts and members of the public, as undertaken by The Tornado and Storm Research Organisation (TORRO) in the UK. The type of data recorded can include thunder heard, lightning seen, thunderstorm cell movement and severe weather observations. Archive data is similar to spotter networks, in that it relies on human observation, but it does not necessarily form part of an organised network and may take many different forms such as academic papers (Gray and Marshall, 1998), newspaper articles and historical diaries (Munzar and Franc, 2003). This kind of data can help verify other observations or extend records back in time but can also suffer from sporadic coverage in both time and space as well as being difficult to consistently gather and classify (Schuster et al., 2005). Satellite and radar technology, where available, are sometimes used in combination with human observations to provide complementary information such as identifying whether observations at different locations are the result of the same thunderstorm (Tippett et al., 2015). Table 1 provides a summary of advantages and disadvantages of manual records for the purposes of compiling lightning and thunderstorm climatologies.

### 2.2 Thunderstorm remote sensing: satellite and radar

Satellite and radar data are often used as a primary source of information for compiling thunderstorm distributions. For satellite sensing, in the absence of additional data to confirm whether convection is thundery, cloud top temperatures are analysed to identify those cold enough to likely be a thunderstorm(Bedka, 2011; Gray and Marshall, 1998). For radar, a thunderstorm is diagnosed by identifying the reflectivity values that are most likely to be attributed to a thunderstorm, examples include 40 dBZ reflectivity value (Haberlie et al., 2016) and 46 dBZ (55 dBZ for a thunderstorm with hail) (Wapler and James, 2015). Diagnosing thunderstorms using satellite and radar data in isolation therefore provides a probable (but not definitive) thunderstorm distribution. Alternative datasets such as ground-based lightning location systems provide absolute confirmation that a convective cloud is a thunderstorm, because lightning is a necessary condition for a thunderstorm (Houston et al., 2015). Lightning information can be used to assess the success of different temperature/reflectivity values in discriminating thunderstorm cells or it can be used in place of temperature or reflectivity values to discriminate thunderstorms

cells that can then be tracked by radar once identified.    Table 2 provides a summary of advantages and disadvantages of remote sensing data for the purposes of compiling lightning and thunderstorm climatologies.

**2.3 Lightning remote sensing: satellite and ground-based lightning location systems**

Lightning location systems were first established several decades ago to collect data on lightning activity. Lightning data quality is primarily assessed by calculating detection efficiency (DE) and location accuracy (LA). Detection efficiency is the percentage of the total number of lightning flashes or strokes a system detects and location accuracy is the median distance error of detected lightning location.  Satellite based lightning location systems detect lightning using an imaging sensor measuring the near infrared spectrum over a large field of view (Nag et al., 2015).  This type of system is thought to have a high detection efficiency relative to ground-based systems (Bitzer et al., 2016) but because, until recently, the satellites detecting lightning have been in a low earth orbit they do not provide continuous temporal coverage, only detecting lightning in an area as the satellite passes over.  They also have a relatively low orbital inclination (near the equator)  which means they do not cover higher latitudes (Thompson et al., 2014).    High earth orbit geostationary satellites in the GOES programme were launched in 2016 and 2017 providing continuous lightning monitoring over the Americas, Pacific and Atlantic oceans (Goodman et al., 2012).  Coverage is a function of instrument range and the areas observable from the instrument's position.

Ground-based systems use sensors to detect the electromagnetic waves that propagate through the atmosphere between the ground and the ionosphere (Hudson et al., 2016). Long-range lightning location systems detect electromagnetic waves in the low and very low frequency range. This is because low frequency waves can travel significant distances (up to 6000km) without significant attenuation  (Said et al., 2010).  The lightning strike location and time are determined by either using their arrival times to calculate the distance travelled or measuring the angle the wave arrives from to triangulate the origin point.  This data can be collected continuously and made available in real-time. Table 3 provides a summary of advantages and disadvantages of lightning remote sensing for the purposes of compiling lightning and thunderstorm climatologies.

**2.4 Thunderstorm indices (proxy data) utilising reanalysis data**

One last dataset type to consider is reanalyses.  Reanalyses use climate data from a large array of sources to model changing climate variables over a long time-period. This provides a consistent spatial and temporal resolution over multiple decades, allowing climate processes to be studied (The Climate Data Guide., 2016).  Reanalysis data has been used in conjunction with other thunderstorm climatologies to identify the synoptic conditions that promote thunderstorm formation or which influence their behaviour in particular regions (Wapler and James, 2015).  The variables used to classify these synoptic conditions into 29 weather patterns were mean-sea-level pressure, geopotential height at 500 hPa, 500-1000 hPa relative thickness and total column precipitable water. Another approach is to calculate average daily values of relevant reanalysis variables such as 500 hPa and 1000 hPa geopotential heights, 500 hPa air temperature and the instability index known as CAPE (Convective Available Potential Energy) for a given temporal resolution (Gatidis et al., 2008).  Reanalyses can also be used to obtain a longer climatology of thunderstorms by developing indices as proxies of thunderstorm activity (Kaltenböck et al., 2009; Kunz, 2007). This can also allow models of future thunderstorm trends to be developed (Tippett et al., 2015).  Different indices may be more or less successful either in general or in different regions and seasons.  An

example of a commonly used index is CAPE which uses two of the three main ingredients for deep moist convection (namely instability and moisture) to evaluate the thunderstorm potential of environmental conditions (Moncreiff and Miller, 1976). The numerical CAPE value indicates the atmospheric potential to produce thunderstorms either looking at current conditions for forecasting or reconstructing the atmospheric conditions of the past for climatology (Holley et al., 2014). Table 4 provides a summary of advantages and disadvantages of thunderstorm indices for the purposes of compiling lightning and thunderstorm climatologies.

Given the variety of datasets and the advantages and disadvantages of each, both the method and the use of data must be carefully considered in light of the overall goal of the research and the characteristics of the study area itself. For example, in Australia some regions are so remote that there are no continuous human thunderstorm observation data making it impossible to achieve a long climatological record using direct observations of thunderstorms (Allen and Karoly, 2014). For the purposes of analysing the effect of ENSO events, a long record is essential, so the method in this event is dictated by the only dataset available in that study area suitable to achieve the goals of the research, namely reanalysis data.

### 3. Thunderstorm Frequency

A wide variety of different methods has been used when creating a climatology of thunderstorms focused on thunderstorm days or thunderstorm frequency. This variation is due to differences in how a thunderstorm day is diagnosed or defined, and how different datasets can be employed in this regard. Figure 3 provides a diagrammatic summary of the different variables to consider during the design of a thunderstorm frequency climatology.

### 3.1 Manual observation

Human observations and archives produce the longest observational record and this enables analysis of long-term trends in occurrence and correlation of thunderstorm frequency with long-term cycles / climate signals such as ENSO (Tippett et al., 2015). Correlations with such cycles may help with the predictability of thunderstorm activity. Pinto (2015) was also able to identify increasing thunderstorm activity in areas of urban heat island development from growing cities in Brazil. In the USA observational records exist for over 100 years and after checking that any variations in the data are not the result of data collection inconsistencies, long term fluctuations demonstrated an overall decrease in thunderstorms over a 40-year period (Changnon, 2001). Nevertheless, this inter-annual variability in thunderstorm activity was found to vary regionally within the USA and six distinct time-series were identified with peaks in activity all occurring in different years and showing a marked difference to the overall national trend. This difference highlights the importance of considering different spatial scales when producing a thunderstorm climatology.

Different studies define thunderstorm days, hours and onset times in alternative ways. For example a thunderstorm day has been defined as thunder heard once in a 24 hour period (Enno et al., 2013) and a thunderstorm is noted to begin when first observed and end 15 minutes after the last thunder is heard (Enno et al., 2013). There is the potential for "false alarms" if there is only one instance of thunder-heard because other noises may be mistaken for thunder. When counting the number of thunderstorms in a day, to ensure that this is done correctly, observations must be separated in time and space (Bielec-Bąkowska, 2003). If thunderstorms start and end on different days consideration should be given to the purpose of the research, if this is to identify the probability of

days with thunderstorms then both days can be counted. However, if the frequency of thunderstorms is of more importance, attributing the thunderstorm to the most appropriate day will avoid night-time thunderstorms being counted twice, inflating thunderstorm-day frequency in those regions.

As shown in Table 1, human observations may contain data from multiple stations, potentially over large areas and in some cases continents, which poses issues with regard to bias and inhomogeneity of data (Schuster et al., 2005; Tuovinen et al., 2009). A European study over a 4-year period utilised records from several different countries and showed that there was likely to be a variable bias due to different data collection techniques (van Delden, 2001). To correct for this the frequency of thunderstorms per 1000 weather reports at each station was calculated in the belief that this would help correct bias incurred by weather stations being manned inconsistently. Other statistical methods used included filling any data gaps using correlation with nearby stations (that show the closest temporal synchronicity) and testing the homogeneity of the data to help choose which stations to be included, and excluding stations which have large data gaps (Enno et al., 2013). The study of Enno produced a climatology of almost 50 years, which showed clear temporal trends, and distributions that could be linked with three main thunderstorm regimes.

### 3.2 Remote sensing: satellite and radar

Radar reflectivity values are used to quantify the severity of convective events including thunderstorms (Tippett et al., 2015) and to diagnose mesoscale convective systems (Punkka and Bister, 2015); catalogue the percentage of thunderstorms that become intense; and identify thunderstorm initiation times and duration (Mohee and Miller, 2010). In Texas, radar was used to establish a link between the presence of man-made reservoirs and thunderstorm initiation, with the caveat that the reflectivity threshold must be sustained for at least 30 minutes (Haberlie et al., 2016). The benefit of radar data over human observation is increased confidence for establishing onset times, geographical extent and precise location of the storm. In contrast, with radar data it can be more difficult to distinguish a thunderstorm from an ordinary convective cell by only measuring precipitation intensity. Some very heavy precipitation is not associated with thunderstorms. Satellite imagery can be used in much the same way as radar to identify thunderstorms because it shows the convective area through cloud presence (Gray and Marshall, 1998); cloud top temperatures below -32°C are used to identify mesoscale convective systems and -52°C used to classify mesoscale convective complexes (a particularly severe form of mesoscale convective system). Severe weather reports associated with thunderstorms have been matched to convective areas in satellite imagery which are significantly colder than the surrounding cloud area and therefore identified as the updraft from deep moist convection (Bedka, 2011).

### 3.3 Remote sensing: satellite and ground-based lightning location systems

Lightning data is commonly used in lightning flash density thunderstorm climatologies. However, there can sometimes be an overlap between lightning flash density and thunderstorm frequency, when lightning data is used to identify thunderstorm days (also referred to as lightning days). A thunderstorm day or lightning day is defined by a certain number of lightning events per day/per area. A reasonable minimum threshold of lightning strikes per area is important because a single strike might be the result of false detection. A successful threshold can be verified with alternative datasets such as human observation and radar; Wapler and James (2015) showed that 2 lightning strokes within a 15km radius was found to be the most effective.

Thunderstorm or lightning days can also be used within a lightning flash density study to establish whether a high lightning area is the result of frequent storms (with attendant high probability of lightning) or less frequent but very intense storms (Soula et al., 2016; Taszarek et al., 2015; Vogt, 2014; Xia et al., 2015). In addition, it can also highlight areas that suffer from frequent thunderstorms which produce only a small amount of lightning, but which may produce other types of hazardous weather such as heavy rain (Xia et al., 2015). It is also useful to ascertain if there are particular regions that favour production of severe thunderstorms (Taszarek et al., 2015). With this in mind, knowing if there are regions that have a lower detection efficiency (percentage of lightning detected by a lightning location system) can be important. This is because whilst detail on storm intensity (number of lightning strikes per storm) is an advantage of lightning data, spatial variations in detection efficiency may bias the results when comparing storms over a large area. Careful validation of results should be undertaken through comparison with other complementary datasets. Also, as lightning location networks have developed more substantially over time, manned thunderstorm observation stations have reduced in number (Enno, 2015) so ascertaining how best to combine manual observations with lightning data may be necessary to maintain a long record. In the USA the two datasets correlate best in areas with high lightning activity (Reap, 2002). For northern Europe it was concluded that the optimum distance for lightning data to correlate with manual records kept by weather stations was in the range 9-14km radius of the observation station depending on the station location (Enno, 2015). It seems that combining two datasets to obtain a long record should be done with caution and the compatibility of the datasets assessed on a case-by-case basis.

Studies use multiple datasets not only to extend the record in time but also to obtain more detail in relation to a thunderstorm climatology. Human observations and records can include details of damage and observations of severe weather events which when compared to lightning data can be used to classify the severity of a thunderstorm (Kaltenböck et al., 2009). It was noted that this approach is only likely to be successful in populated areas where severe weather and damage was more likely to be recorded and observed.

**3.4 Thunderstorm indices (proxy data) utilising reanalysis data**

Reanalyses, such as ERA5 European Reanalysis data, have assimilated observational records of land, ocean and atmospheric variables into models from a large variety of observational sources since 1979 and in 2020 will have extended the record back to 1950 (Hersbach et al., 2019) have been employed to identify the atmospheric conditions common to regions and seasons of high thunderstorm activity. This does not produce a thunderstorm frequency climatology because there are no direct records of thunderstorm activity. However, they can produce a frequency of thunderstorm promoting conditions. In Australia, reanalysis data was used to reconstruct a climatology of the atmospheric environment conducive to the development of severe thunderstorms (Allen and Karoly, 2014). This insight assists forecasters in identifying the conditions that have a high probability of generating a hazardous thunderstorm. Indices such as CAPE or LI (lifted index) can be used to predict thunderstorm occurrence based on the atmospheric conditions and if generated from reanalysis data then a long record can be produced of the potential for thunderstorm formation, which should ideally then be ground-truthed against measurement data. In Southwest Germany different indices were tested against severe thunderstorms identified in SYNOP weather station data, radar data and damage reports to ascertain which index or indices work(s) best in which scenarios (Kunz, 2007). This has also been done on a continental scale for the whole of Europe using lightning location system data, severe storm reports and weather forecast model output data to verify

the degree to which indices can reliably predict thunderstorms (Kaltenböck et al., 2009). In the USA reanalysis data and indices were used to identify conditions with a high probability of producing severe thunderstorms (defined by hail size, gust speed or tornado damage) (Brooks et al., 2003). These findings were then applied to Europe to produce a climatology of conditions which have the highest probability of producing severe thunderstorms. The results agreed with thunderstorm frequency work that has been done in Europe however, without a long-term Europe wide climatology the success of this approach remains uncertain.

## 4. Thunderstorm Tracking

Another useful approach is reconstruction of thunderstorm tracks, recording thunderstorm movement which is typical in a specific region, synoptic pattern or time-period (season, time of day, month etc.). This might include data such as thunderstorm lifecycle duration (an individual cell or multi-cell thunderstorm), direction of travel, speed, development of intensity (such as lightning or rainfall hazards throughout the life of the storm) and can also include a form of thunderstorm frequency (how often a thunderstorm tracks through a particular area) (Galanaki et al., 2018; Gray and Marshall, 1998). This type of information can help forecasters to identify areas at risk from thunderstorm hazards or assist with now-casting (predicting the movement of an existing storm based on the previous trajectory of the cell), or general climatology. Figure 3 provides a diagrammatic summary of the different variables to consider during the design stage of thunderstorm tracking research.

### 4.1 Manual observations

Tracking may be possible using manual observations and archive information but it is problematic to connect thunderstorms from one observation location to another and to confidently identify them as the same storm. Therefore this data is often used in combination with other datasets such as satellite and radar (Gray and Marshall, 1998). This study enabled the reconstruction of mesoscale convective system (MCS) tracks over a 16-year period in the UK. An MCS is a collection of thunderstorm cells which make up a continuous storm area that extends over 100km in at least one direction (Doe, 2016). The benefit of using this combined dataset in this case was that as the UK experiences infrequent MCS's a long period was required to obtain enough tracks for a climatology and the human observations provided confirmation that satellite and radar data diagnosis of a thunderstorm occurrence is correct. This was later updated for a further 17 year period (Lewis and Gray, 2010) to provide a database of MCS tracks for a total of 23 years for the UK. The climatology is used to identify trends in origin points for storms, duration, start and end times of storms and to link trends in behaviour to specific synoptic conditions. In this case, inclusion of satellite and radar provided additional confidence, but it was noted that some MCS may have been diagnosed incorrectly because where only human reports were available, multiple but separate scattered thunderstorms may produce a similar distribution of reports to an MCS.

### 4.2 Remote sensing: satellite imagery and radar

Radar and satellite imagery are often used to track thunderstorm cells in real-time for the purpose of nowcasting (anticipating the next most likely movement of the cell) using 3d reflectivity profiles to define the extent and structure of a thunderstorm (Dixon and Wiener, 1993; Johnson et al., 1998; del Moral et al., 2018). These tracking algorithms have also been applied to historical thunderstorms to develop a catalogue of thunderstorm movements and severity (Chronis et al., 2015; Farnell and Rigo, 2020). Radar tracked thunderstorm data can be used by

industry responsible for infrastructure such as power lines to develop risk models (Mohee and Miller, 2010) and enhance resilience. Detecting thunderstorms at longer ranges is challenging for radar, a problem which can be overcome by using multiple radar devices (Mohee and Miller, 2010). When using output from multiple radar datasets they need to be merged into a composite so that thunderstorm clusters can be tracked (Lock and Houston, 2015). The linking of clusters into a track has been achieved both using wind direction data (Lock and Houston, 2015) and the previous motion of the storm (Dixon and Wiener, 1993; Johnson et al., 1998; del Moral et al., 2018). The initiation point of a thunderstorm can be approximated by interpolating backwards using the trajectory of the thunderstorm by a time step of 15 minutes before it was first detected (Lock and Houston, 2015). This can be useful because it can take thunderstorms time to develop to the point where the reflectivity is high enough to be detected and the first detection by radar is not necessarily representative of the start location for the storm.

There may also be a similar detection delay using satellite data, as this is usually only available every 15 minutes so there is a potential for 15-minute error windows for start and end times (Dotzek and Forster, 2011). Finding the origin point for the storm assists in identifying the conditions that contribute to their formation and, in this case, in correlating thunderstorm formation hot spots with topography as well as identifying the overall spatial distribution of thunderstorm formation.

Radar reflectivity values for thunderstorm tracks can also be used to provide information on severity of thunderstorm precipitation and to quantify how this changes as the storm develops and dissipates (Rigo and Pineda, 2016).

**4.3 Remote sensing: satellite and ground-based lightning location systems**

Thunderstorm intensity changes have also been inferred from lightning activity (Correoso et al., 2006) by analysing the lightning intensity per 100km$^2$ for each 30 minute stage of the life cycle of 33 MCSs. It was noted that colder storms and the early stages of storms produced the most lightning. There have been numerous studies (Chronis et al., 2015; Farnell and Rigo, 2020; Schultz et al., 2009) that have identified a "jump" in lightning activity within a thunderstorm (e.g. 2 standard deviations above the running mean of lightning strokes from the previous 12 minute iteration) as a means of identifying storms which can be tied to observations of severe weather. Research in this area is ongoing to establish how a warning system based on lightning intensity can be adapted to different regions, which may produce different patterns of thunderstorm activity (Ellis and Miller, 2016) and identifying the best combination of the variables to produce the highest probability of detection whilst maintaining a low false alarm rate(Gatlin and Goodman, 2010).

Lightning data has also has been used for thunderstorm tracking purposes either with or without supporting information from radar, satellite and human observation. The main decisions when using lightning data for tracking are a) deciding how to define a lightning cluster so that it most closely represents the thunderstorm cell or thunderstorm as a whole, and b) how to connect the clusters to produce an accurate track. Identifying a cluster usually involves counting lightning strikes within a given time interval and within a given radius or grid square, the method for doing so varies depending on whether the study aims to track individual thunderstorm cells or whole thunderstorms (which may include multiple cells). For example, a radius of 10km and 16 minutes time interval was chosen (around each lightning strike) as a means of counting strikes that originate from the same storm in a study in the Mediterranean region (Galanaki et al., 2018). These parameters compared well with

satellite imagery showing the cloud extent. In another study undertaken in the Alps a thunderstorm cluster was defined as a minimum of 14 flashes within a 4km radius and 20-minute temporal vicinity. Lightning flashes that did not meet this requirement were discarded because this study wished to exclude "weak storms" from the dataset (Bertram and Mayr, 2004). The difference in size is likely a function of differing thunderstorm activity or size between the study areas, which is also therefore an important consideration when choosing cluster size. Other important considerations for cluster size may be the maximum distance a lightning strike can travel from the convective core and the detection efficiency or location accuracy of the dataset itself.

As with satellite and radar data, connecting the lightning clusters into a track can be challenging because there can be multiple thunderstorm cells or multiple thunderstorms in a similar area (which can also split and merge) (del Moral et al., 2018). Tackling this problem has been addressed in a variety of ways. Identifying the mean wind direction between 0 to 6km elevation (Houston et al., 2015) and choosing the lightning cluster that most closely matches the trajectory of the gradient wind is one method. It should also be noted that some thunderstorms are large enough to move deviant from the flow (del Moral et al., 2018). A different approach was employed in the Alps specifying that clusters could be connected within a 30 degree +/- direction variation of the mean cell motion of that region (Bertram and Mayr, 2004). This required initial data analysis prior to track construction to calculate the mean by connecting cells that are closest to each other over a whole day period and gathering data for direction and distance of movement. For unusual flow situations the direction can be changed to avoid incorrect tracking (the process is semi-automated to allow this). Lastly, another method of connecting clusters into a track is ensuring that the time iterations are small enough to provide a spatial overlap (Meyer et al., 2013).

Some problems with using lightning to track thunderstorms include the fact that lightning may not begin at the convective start of the storm, making the initiation point uncertain and there is also difficulty detecting cloud based lightning which is the dominant lightning type for early thunderstorm stages (Bertram and Mayr, 2004). Thunderstorms that are less electrically active may escape detection.

**5. Lightning flash density**

Lightning flash density studies use data from lightning location systems and some standardised analysis methods of best practice have been developed when using these datasets. Whilst most lightning climatologies are produced with the intention of minimising exposure to cloud-to-ground lightning hazards (Finke 1999), lightning climatology can also be viewed as a form of thunderstorm climatology because lightning can be used to confirm thunderstorm activity. Indeed, there are several avenues of research investigating how lightning might be used as a proxy for other thunderstorm hazards such as heavy precipitation (Ezcurra et al., 2002; Iordanidou et al., 2016; Kochtubajda et al., 2013). Lightning flash density studies can overlap with thunderstorm frequency studies when they include "days with lightning" as part of the climatology.

Whilst high lightning flash density may provide an indication of increased thunderstorm activity this should be treated with caution because it may not so easily detect low lightning thunderstorms, which while less electrically active, may still produce other forms of hazardous weather. This may be remedied by analysing thunderstorm or lightning days (see section 3) in conjunction with lightning flash density. Lightning flash density information can support understanding of lightning and thunderstorm distributions amongst industry end-users. Ground flash density (Diendorfer, 2008) is used to calculate the risk from lightning to an asset, and is relevant to operations

such as wind farms, shipping and sailing, sporting events and transport infrastructure, as well as many other types of industry and outdoor land use, especially where cloud to ground lightning poses a hazard to life. Figure 4 provides a diagrammatic summary of the steps involved in producing a lightning flash density (thunderstorm) climatology and the different variables to consider during study design.

## 5.1 Lightning flash density method


Whilst thunderstorm frequency uses different types of datasets and different methods, lightning flash density studies depend upon a variety of lightning datasets (lightning location systems vary in detection method, coverage and accuracy). However, they usually follow a relatively standardised methodology, making results easier to compare. Most studies focus on cloud to ground lightning because they are primarily concerned with lightning

strike damage, but also because most ground-based lightning location systems detect cloud to ground strikes most efficiently. These studies often have a shorter timescale than most other climatologies because lightning location networks experience upgrades that limit the period over which they are homogenous. Some systems operate over a limited timespan (Tropical Rainfall Measuring Mission Optical Transient Detector for example: (Cecil et al., 2014). For lightning detectors placed on satellites, data collection is limited by the satellite deployment duration.

Where lightning flash density is required for industry purposes (to obtain a lightning flash density figure as input, for example, to risk assessment models for construction) but no lightning flash density is available, it has been estimated by multiplying days of thunder heard by 0.1 (DEHN, 2014). Whether this calculation can be used successfully to convert a long record of days with thunder to lightning flash density, where human observations have been replaced by lightning location systems, to produce a long climatology record remains to be seen.

Data often needs to be filtered to omit weak events which may not be the result of cloud to ground lightning, and individual lightning strokes need to be grouped into lightning flashes (Taszarek et al., 2015). The threshold for excluding weak events may differ depending on the dataset, coverage area and purpose of the study (some may wish to exclude cloud-to-cloud lightning events). Grouping of lightning strokes into flashes is performed by setting an arbitrary time-period and spatial area within which if strokes occur together, they are almost certainly

the result of the same lightning event. Most studies follow the definition that a flash is an ensemble of all strokes within 10km of each other within a one-second interval (Cummins and Murphy, 2009). It is noted that the temporal element of this is the most important, with one second being consistent throughout the literature but the spatial element is more variable (Drüe et al., 2007) as it does not appear to significantly affect the number of grouped flashes, even up to as much as 50km.

Consideration should also be given to network upgrades, which may affect detection efficiency. Some studies choose timescales and locations which do not include a significant upgrade to obtain homogenous data (Taszarek et al., 2015) while others apply corrections to homogenise the time series (Huffines and Orville, 1999). Applying corrections may provide a longer timescale for a study than would otherwise be possible. Using longer time series is usually more reliable because it minimises the influence of some biases, such as sensor outages or unusually

severe weather events. However, choosing a known homogenous data collection period may be the safer way forward, even if it limits the length of record available.

Lightning flash density per km² per year is usually calculated throughout the study area on a grid square basis. The grid box should not be smaller than that required to capture a minimum of 80 lightning events (Diendorfer, 2008) to provide an 80% confidence that the calculated ground flash density is an accurate representation. Adjustments to ensure that there are 80 events per grid cell may be either a function of grid box size or study duration. For a location accuracy that is between 500m to 1000m the grid size should be no smaller than 1km x 1km (Diendorfer, 2008). The size of the grid box may also vary depending on the size of the study region and the resolution required to address the research question. One suggested improvement for this is to use probabilistic methods to obtain a sub km lightning flash density resolution which would be better suited to analysing the relationship between lightning and smaller scale landscape- and biological features such as vegetation (Etherington and Perry, 2017). It has been shown to be possible to produce a 100m x 100m climatology by calculating the radius around a lightning location, that it is most probable that the strike occurred within, using the known location error data from the lightning location system. The probability of a strike occurring within an area of interest can then be calculated. This method produces a detailed map, however the extra processing required makes this method unlikely to be adopted as standard practice.

Once an appropriate grid size is identified, flash density can be calculated per km² per year for each grid box. Temporal and spatial variations of lightning flash density are then analysed and can include investigations of the impact of potential influencing factors such as topographic features, land use, CAPE(Galanaki et al., 2015), synoptic conditions (Gatidis et al., 2018) and aerosols (Coquillat et al., 2013).

**5.2 Global lightning flash density**

An advantage of lightning location system data is that some systems operate over very large areas allowing lightning flash density to be analysed on a global scale. A comparison study was produced, using both a ground based lightning location system (the World Wide Lightning Location Network WWLLN) and a satellite based system (TRMM OTD and LIS), to ascertain whether the lower detection efficiency of WWLLN had consequences for its identification of diurnal cycles (Virts et al., 2013). The results showed that WWLLN was able to produce plausible diurnal cycles on a regional and global scale. Both datasets picked up the general trends of geographical and seasonal lightning variation but there were areas where one dataset would detect greater lightning amounts than the other (OTD/LIS detecting more lightning in Africa and the Himalayas vs WWLLN detecting more over the oceans) reflecting the fact that each lightning location system's performance varies spatially.

Unsurprisingly, global maps of lightning flash density show most intense lightning activity in the tropics due to the intense solar heating initiating convection. Mountain ranges often show greater lightning activity than their surrounding areas (Cecil et al., 2014) due to sun-facing slopes and forced ascent of air helping to release instability. Lightning hot spots have been ranked and vicinity to populated areas recorded to highlight areas that experience high lightning risk and which are more vulnerable to thunderstorm and lightning hazards (Albrecht et al., 2016). Further studies of vulnerability and lightning flash density could usefully include recreational areas, areas with high risk activities and infrastructure.

**5.3 Lightning flash density and topography**

Strong correlations between mountain ranges and enhanced lightning activity (in comparison to lightning intensity in surrounding lowlands) are noted in numerous global studies (Etherington & Perry, 2017; Feudale & Manzato,

2014; Mushtaq et al., 2018; Vogt, 2014; Vogt & Hodanish, 2014, 2016; Xia et al., 2015). More analytical information can be obtained by attributing a mean slope or elevation value to each grid square (Galanaki et al., 2015) and choosing appropriate statistical methods to establish correlation. Another method is to create Shapefiles in a GIS environment for each elevation class and to calculate the lightning flash density for each (Vogt and Hodanish, 2016) or join shape files containing elevation data to a lightning density grid to obtain elevation data

for each grid cell environment (Mushtaq et al., 2018). Slope gradient is another element of topography that may influence lightning flash density, for example in Colorado where it was noted that lightning flash density increases more rapidly at higher elevations (steeper slope gradients) than at lower elevations (gentler gradients) (Vogt and Hodanish, 2014).

**5.4 Lightning flash density and aerosols**

There have been several studies examining the influence of aerosols on lightning flash density. Comparing lightning activity during the week with weekend days around commuter/urbanised areas, anthropogenic emissions (during the week) were shown to increase the intensity of lightning activity downwind of Paris because at weekends the lightning activity was less intense (Coquillat et al., 2013). It is argued that natural causes would not change from weekdays to weekends. On a longer timescale, an alternative approach obtained monthly averages

of the absorbing aerosol index for each flash density grid cell and calculated the correlation between this and lightning flash density in the Kashmir and Jammu provinces of India. A positive correlation (r=0.61) identified that aerosols may be an influencing factor in controlling lightning activity in these regions (Mushtaq et al., 2018). Urban heat island temperature has been observed to exhibit a maximum on Fridays and minimum on the weekend. In the Charlotte, North Carolina urban heat island it has been observed that there is a slightly higher mean

temperature (1°C) on weekdays to weekend days (Eastin et al., 2018). Increased temperature during the week may therefore also be a factor influencing increased lightning activity.

**5.5 Lightning flash density and land cover**

Evaluating the connection between land use/vegetation type and lightning can depend on available datasets. This requires the classification of regions or obtaining land cover classification datasets and attributing this

classification to the lightning flash density grid square (Galanaki et al., 2015), or calculating lightning flash density stratified by land use polygons per season. The relationship for an area can then be quantified by scaling the lightning stroke density with the total number of strokes and percentage area of each vegetation/land use category to the total study area. An analysis for different vegetation types in the Eastern Mediterranean region (Galanaki et al., 2015) showed that seasonal variation of lightning activity varied between them. For example, in summer

lightning showed a preference for forested areas thought to be the result of greater soil moisture and leaf areas permitting more transpiration of moisture into the air. Scrubland showed low lightning activity throughout the year and in the coldest periods of the year there was increased lightning activity in woodland and wooded grassland.

**5.6 Lightning flash density and atmospheric conditions**

Correlating lightning activity with meteorological, synoptic or local atmospheric conditions is important to understand how this may affect the distribution of lightning and thunderstorm related hazards. Analysis of the influence of atmospheric conditions is often undertaken using reanalysis data (e.g. Gatidis et al., 2018). Using

factor analysis for lightning flash density across Greece in fortnightly time iterations for each 0.5-degree grid square, this study was able to identify three main intra-annual distributions of lightning activity. Namely, high

activity occurring in a) continental mountainous areas in early summer, b) over the Ionian Sea in early autumn, and c) over the Aegean Sea in late May and again in mid-autumn. Once the temporal and spatial distributions of the three main peaks in lightning activity were identified, mean atmospheric conditions (average patterns of geopotential heights at 500 hPa and 1000 hPa, air temperature at 500hPa and CAPE) were obtained on days where there was lightning activity during the peak "season" of activity for each case. This allowed the identification of

the atmospheric conditions that were most strongly associated with the lightning activity. The benefit of using factor analysis for fortnightly time periods, rather than a traditional seasonal/monthly analysis, is that it removes the possibility that by parcelling time by human constructs (i.e. months) critical transitions may be missed. Factor analysis ensures objective grouping to identify the main trends (Gatidis et al., 2018).

Thunderstorm indices such as CAPE have been widely evaluated in conjunction with lightning flash density

(Galanaki et al., 2015). Convective Available Potential Energy quantifies the atmospheric conditions' potential for deep moist convection. Galanaki et al (2015) assigned CAPE values into bins for several times of day and then the lightning activity for each time of day was paired to the corresponding CAPE bin. The results show an increase of lightning activity with increasing CAPE values, with a positive correlation of $R > 0.87$.

Research can also include the effects of long term variations in atmospheric circulation, such as ENSO events and

the North Atlantic Oscillation (NAO) (Piper and Kunz, 2017), on thunderstorm day distributions. Lightning activity for ENSO Neutral months can be compared to months with El Niño and La Niña events. This has been addressed in the Northwest Pacific region (Zhang et al., 2018). Abnormal lightning activities were identified during both El Niño and La Niña events. Overall, it was found that there was a 10.3% increase (4.8 % decrease) in lightning days during El Niño (La Niña) events.

**6. Recommendations**

In order to gain the most comprehensive understanding of the distribution of thunderstorm hazards the following recommendations should be considered:

**6.1 Dataset choice**

A major consideration when choosing appropriate underpinning datasets is identifying both the availability for

the study region concerned and the appropriate temporal and spatial coverage required to achieve the overall research goal. Once potential datasets are identified, investigations should ascertain the reliability of the data and the homogeneity of the recording methods (Schuster et al., 2005). The research project may need to be adapted if dataset limitations constrain the types of analysis that can be performed. For example, in a region where only lightning data is available, a short record length may mean that long term trends cannot be analysed and the focus

may need to be on the spatial variation of lightning and thunderstorm occurrence.

Since no one dataset is perfect, it can be beneficial to combine complementary datasets to fill data gaps, validate thunderstorm diagnoses (Gray and Marshall, 1998) and extend spatial and temporal coverage (Enno, 2015). Where datasets cannot be confidently combined, repeating the analysis with more than one dataset can provide validation of results or help to identify the main potential sources of uncertainty.

### 6.2 The benefits of combining different types of approach

There is substantial benefit to incorporating more than one research methodology into a study (thunderstorm frequency, thunderstorm tracking, and lightning flash density) to produce robust results. Good examples of this include lightning flash density climatologies which have incorporated aspects of thunderstorm frequency research (e.g. Soula et al., 2016) since not all thunderstorms produce the same amount or form of electrical activity. Thunderstorm frequency can help distinguish regions that are at risk from rare severe storms from those at risk from frequent less severe storms. Furthermore, differences between thunderstorm frequency and lightning flash density may help identify instances where the spatial variation of lightning flash density has been skewed by severe storms as demonstrated by Anderson and Klugmann (2014).

Thunderstorm frequency and lightning flash density studies can provide data relating to thunderstorm hazard distributions in a fixed region during a fixed period of time but they cannot provide data relating to the movement of thunderstorms. Factors such as storm location origin, thunderstorm lifecycle- and motion characteristics also provide important information to characterise the potential hazard in a region. It is important to investigate both Eulerian and Lagrangian approaches to thunderstorm distributions to fully understand the risk from thunderstorm hazards and identify causative factors such as atmospheric conditions. Lastly, lightning flash density approaches can be used within thunderstorm tracking to see how lightning flash density changes throughout the lifecycle of the storm (Correoso et al., 2006), identifying whether particular thunderstorm types produce more or fewer lightning hazards.

### 6.3 Identify the end-user

Aside from scientific interest, potential end-users should be considered, as this will also influence the choice of method and aim of the research. The study may take the form of analysing hazards for a specific group such as forecasters or nowcasters, mountaineers and outdoor leisure users (Vogt, 2014), or a specific industry such as the power sector (Mohee and Miller, 2010), or more general users of warning services amongst the general public. Identifying the target audience is crucial for tailoring the results so that they can be successfully utilised to mitigate the effects of thunderstorm hazards.

The end-user will also determine how best to communicate the results both in terms of dissemination pathways and presentation format. Weather advice services, warning services and forecasters will access the results via scientific journal articles, conference papers, presentations and training courses. If the study has been produced for a specific organisation then they may also require tables of results or maps which they may interrogate and apply for their own purposes and integrate into their own decision support systems. Decision makers in industry and government, as well as the general public, will require clear diagrams, summaries and guidance on how to interpret the results. In recent years apps, social media posts and websites have become popular with interested members of the public being able to observe lightning strikes and radar imagery in real time and sign up to receive alerts via social media with regard to weather warnings. Utilising such platforms to deliver information in relation to past hazard distributions and developing apps and websites to do so could provide easy access to information for the public and could be a potential growth area to enable climatologists to distribute the results of their research.

### 7. Conclusions – priorities for further research

### 7.1 Low-lightning areas

Research is most often conducted in populated areas of frequent thunderstorm activity, partly because these
regions are more at risk from thunderstorm hazards and partly due to enhanced monitoring producing the
observational evidence to support more statistically significant and reliable results. In areas which experience
fewer thunderstorms, accessing sufficient data to produce statistically significant results or high-resolution spatial
distributions can be problematic. For example, producing a lightning flash density map with 80% confidence
level requires a grid square to have accumulated at least 80 lightning flashes during the study period (Diendorfer,
2008). In low lightning activity areas, to obtain a reasonable sample often requires increasing the grid size or the
time scale, thus potentially limiting investigations into intra-annual and monthly distributions at high spatial
resolutions.

### 7.2 Dataset combination techniques

More accurate thunderstorm distributions can be achieved by enabling more accurate syntheses of different data
sources. This could take the form of developing methodologies and algorithms which support integration and
which can be adapted to incorporate different data types, or alternatively by combining data sets of the same type
such as lightning data from multiple systems.

### 7.3 Reanalysis indices

Testing and improving techniques to define indices from reanalyses could provide a long record of probable
thunderstorm activity, in regions where records are short or inhomogeneous, as well as being used in areas where
there is a lack of thunderstorm observational data. Testing the results against direct observations and identifying
the indices which work best in different regions and seasons would increase confidence in utilising this method.

### 7.4 Hazard communication and warnings

Developing pathways to communicate thunderstorm distributions to lay persons or targeted end-users is necessary
to help them plan in advance to better avoid or prepare for thunderstorm hazards. Apps and social media provide
platforms which are popular and familiar for lay persons, many people now being familiar with real-time lightning
websites and radar imagery. Thus, such methods need to be employed more widely to display climatological data
in a user-friendly way.

**Data availability.** No data was used in the compilation of this paper

**Author contributions.** LH conducted the review of the available literature and wrote the manuscript with MW,
NP and SD assisting with conceptual development, contributions to text and editing.

**Competing interests.** The authors declare that they have no conflict of interest.

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

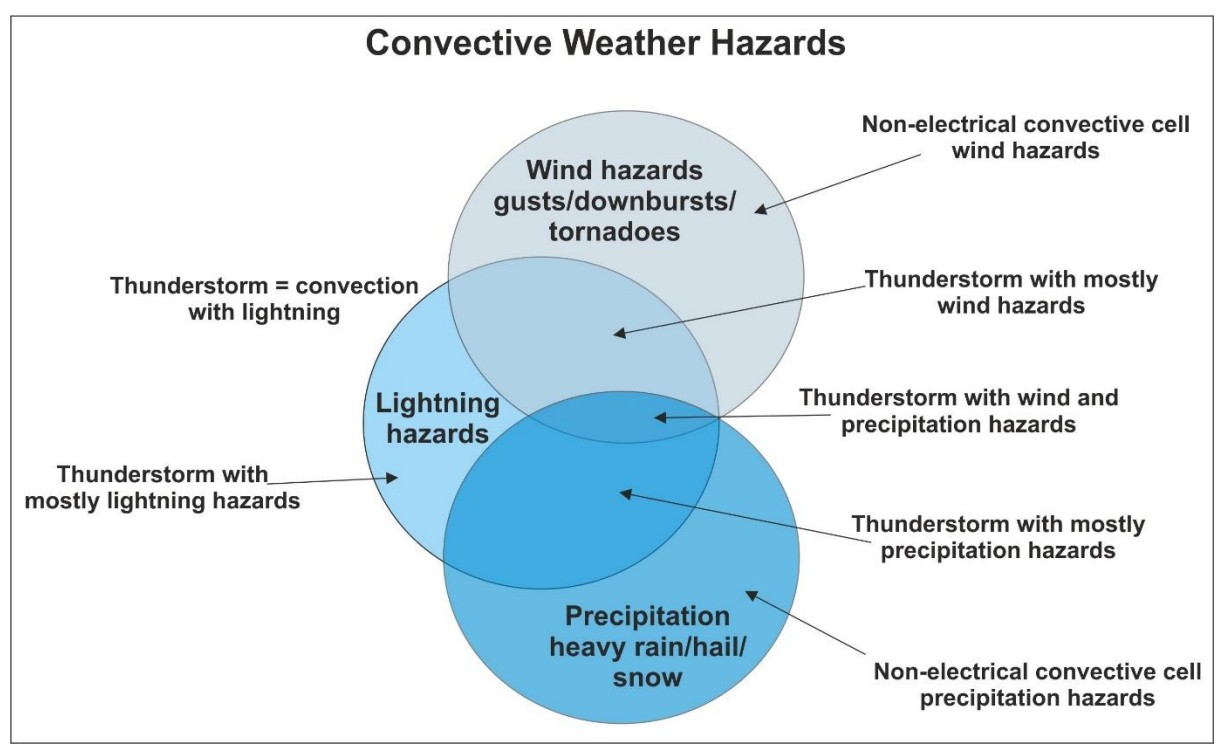

**Figure 1: Venn diagram of the relationship between convective weather hazards and how thunderstorms are distinguished from ordinary convection by electrical hazards.**

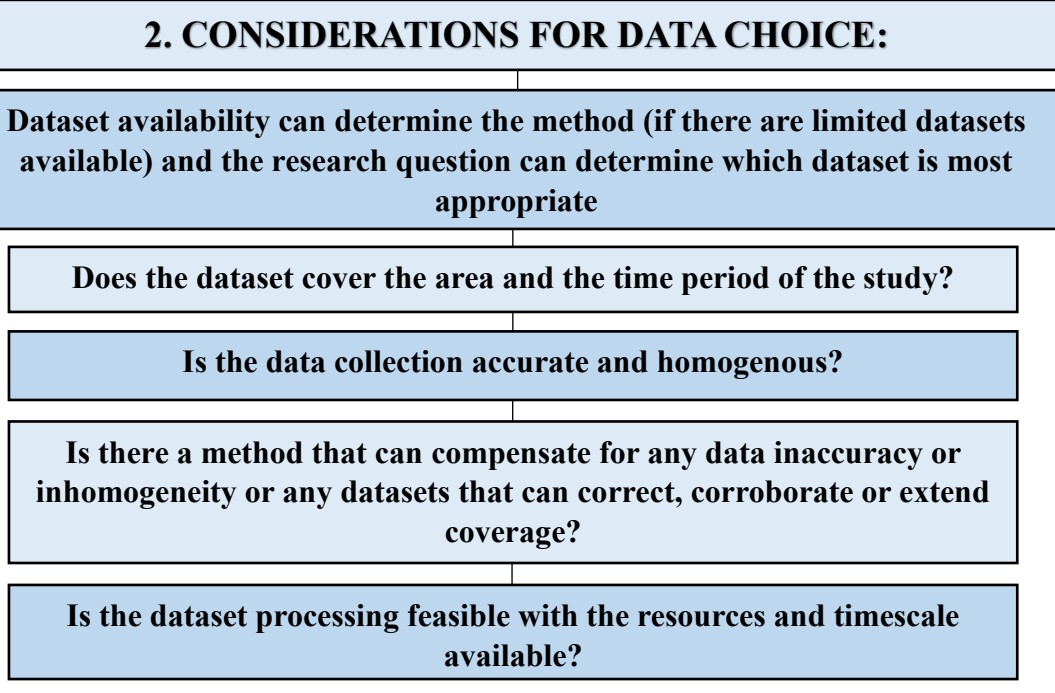

**Figure 2: Checklist of questions to consider when choosing the most appropriate dataset**.

| 2.1. Manual observation data: spotter networks, archives and records | |
| --- | --- |
| Advantages | Disadvantages |
| ➢ Detailed information in relation to storm activity and behaviour (Enno et al., 2013) | ➢ Inconsistent observation and recording methods (van Delden, 2001) |
| ➢ Often provides a long record, in some cases over 100 years(Changnon, 2001) | ➢ Station relocation (Changnon, 2001) |
| ➢ Long record can allow assessment of long term temporal trends and correlation with cycles such as ENSO events (Pinto, 2015) | ➢ Inconsistencies between different station locations ability to hear thunder and see lightning such as topographic barriers, urban areas light and noise interference (Enno, 2015). This may result in one location being able to detect thunderstorms at a much further distance than others. |
| Mitigation for disadvantages: | ➢ Thunderstorms are much easier to observe during the night-time (Enno et al., 2013) |
| ➢ Careful selection of time period and stations used (checking for changes in data collection) (Pinto, 2015) | ➢ Data collection may not be continuous due to absences, holidays, staff shortages and political fluctuations (Bielec-Bąkowska, 2003). |
| ➢ Performing homogeneity tests to the data to see whether practice changes effect the results (Enno et al., 2013) | |
| ➢ Compensating for bias by calculating thunderstorms per 1000 weather observations (van Delden, 2001) | |
| ➢ Checking distribution against other data collection techniques to see if they agree (Wapler and James, 2015) | |

**Table 1: Strengths and weaknesses of manual observations used to produce thunderstorm and lightning climatologies.**

| 2.2 Thunderstorm remote sensing: satellite and radar | |
|---|---|
| Advantages | Disadvantages |
| <ul><li>Shows the spatial extent of the thunderstorm's convective area</li><li>Can sometimes detect thunderstorms with low electrical activity or before lightning activity begins, both of which might be missed by lightning location systems</li></ul>Mitigation for disadvantages:<ul><li>Integration of datasets such as lightning data (Houston et al., 2015), records and spotter networks can be used to confirm diagnosis of thundery activity (Gray and Marshall, 1998) and correct for time error</li></ul> | <ul><li>Identification of thunderstorms is based on reflectivity and cloud top temperatures that are likely to produce thundery activity and does not provide absolute confirmation of diagnosis (Houston et al., 2015)</li><li>Measurements and images are often taken at fixed time intervals so there is a potential error for start and end time of storms (Dotzek and Forster, 2011)</li><li>Satellite imagery can have visibility difficulty for night-time storms (does not affect radar or satellite infra-red)</li></ul> |

**Table 2: Strengths and weaknesses of thunderstorm remote sensing (satellite and radar) used to produce thunderstorm and lightning climatologies.**

## 2.3 Lightning remote sensing: satellite and ground based

| Advantages | Disadvantages |
|---|---|
| ➢ Can detect lightning up to a global scale (Thompson et al., 2014), available in real time | ➢ Satellite systems which are orbital do not provide continuous coverage (Thompson et al., 2014) |
| ➢ Can provide continuous coverage (Vogt, 2014) | ➢ Detection efficiency can vary spatially and diurnally (Poelman, et al., 2013)(Bennett et al., 2010) |
| ➢ Variety of applications ranging from thunderstorm intensity and tracking, to warning systems (Poelman et al., 2013). Can be used as a proxy for other thunderstorm severe weather types | ➢ Can make false detections (Nag et al., 2015) |
| | ➢ Absolute detection efficiency and location accuracy is difficult to establish for whole coverage area (Poelman et al., 2013) |
| ➢ Provides large amounts of data | ➢ Upgrades and improvements to algorithms mean that detection efficiency, false alarm rate and location accuracy may vary over time (Keogh et al., 2006) |
| **Mitigation for disadvantages** | |
| ➢ Choosing a study area and lightning system to ensure homogenous spatial coverage (Bertram and Mayr, 2004) | ➢ Variation of detection efficiency for cloud ground and cloud-based lightning (Betz et al., 2009). Some systems can detect a larger amount of cloud based lightning while others only detect a small amount and are unable to accurately distinguish cloud based lightning from cloud to ground lightning |
| ➢ Choosing a study duration which should have homogenous coverage (Galanaki et al., 2015) | |
| ➢ Carry out corrections for inhomogeneity, detection efficiency or location accuracy (Etherington and Perry, 2017) | |
| ➢ Excluding weak lightning signals that may not be the result of lightning or a false detection (Taszarek et al., 2015) | |

**Table 3: Strengths and weaknesses of lightning remote sensing (satellite and ground-based) used to produce thunderstorm and lightning climatologies.**

| 2.4 Thunderstorm indices (proxy data) utilising reanalysis data | |
|---|---|
| Advantages | Disadvantages |
| ➢ Reanalysis provides consistent spatial and temporal resolution over a multi-decadal time span (The Climate Data Guide., 2016) e.g. ERA5 1979 to date includes many atmospheric, land and oceanic climate variables<br><br>➢ Can help to reconstruct climatic conditions which produce thunderstorms (Allen and Karoly, 2014)<br><br>➢ Can produce longer climatologies (Brooks et al., 2003)<br><br>➢ Can be used to reconstruct thunderstorm activity in areas of poor coverage (Allen and Karoly, 2014) | ➢ Original datasets such as SYNOP surface pressure, temperature, wind and humidity along with a vast array of other datasets are used as input to reanalysis and can vary in collection method, contain biases or not be homogenous (The Climate Data Guide., 2016)<br><br>➢ Using indices provides probable thunderstorm occurrence not direct observation (Kaltenböck et al., 2009)<br><br>➢ Indices may be more or less successful by regions, times of the year and under different climatic conditions (Kunz, 2007) |

**Table 4: Strengths and weaknesses of proxy datasets used to produce thunderstorm and lightning climatologies.**

## THUNDERSTORM CLIMATOLOGY

**3.1 Manual observations:**
- Correct / check inhomogeneity of data
- Define thunder day e.g. minimum threshold of observations per 24 hours / per area
- Correlate thunderstorm frequency with long term cycles and identify long term trends

**3. Thunderstorm frequency:**
- Thunderstorm / lightning days
- Spatial and temporal trends
- Duration of thundery conditions
- Link to synoptic conditions

**4. Thunderstorm tracking:**
- Direction, duration and speed statistics
- Initiations points
- Activity variation throughout
- Correlation of track behaviour to synoptic conditions

**4.1 Manual observation:**
- Observe storm movement
- Record arrival and departure times
- Difficult to connect observations at separate locations into a confirmed thunderstorm track
- Recommend use only with another dataset

**3.2 Thunderstorm remote sensing:**
**Radar** data thunderstorm classification: minimum reflectivity value
**Satellite** data thunderstorm classification: cloud top temperatures:
- Minimum time this value must be maintained to be classified as a thunderstorm

**3.3 Lightning remote sensing:**
- Thunder day: minimum strikes per day / per area
- Check detection efficiency homogeneity for study area
- Distinguish regions that experience different thunderstorm frequency and different lightning intensity

**3.4 Thunderstorm indices (proxy data) utilizing reanalysis data:**
- Ascertain which indices is most successful (varies spatially and for different atmospheric conditions
- Use reanalysis data with chosen indices to produce a climatology of thunderstorm producing environments
- Reanalysis data can also be used with results of direct observation thunderstorm climatology to identify the atmospheric conditions common to high thunderstorm activity

**4.3 Lightning remote sensing:**
- Thunderstorm cell defined lightning cluster with minimum strikes per time step / per area
- Linking clusters by spatial crossover / wind direction / average movement by region
- Lightning intensity monitoring as severe storm indicator

**4.2 Thunderstorm remote sensing:**
**Radar** data track identification: minimum reflectivity value may require data from multiple radars
**Satellite** data track identification: cloud top temperatures / cloud extent
- Linking cells at next time step by previous motion / wind data or both
- Backwards interpolation to identify initiation point due to delay in thunderstorm detection
- Thunderstorm intensity monitoring

**Figure 3: Diagrammatic summary of the potential research findings and data utilisation for a thunderstorm climatology created using either thunderstorm frequency or thunderstorm tracking methodologies.**

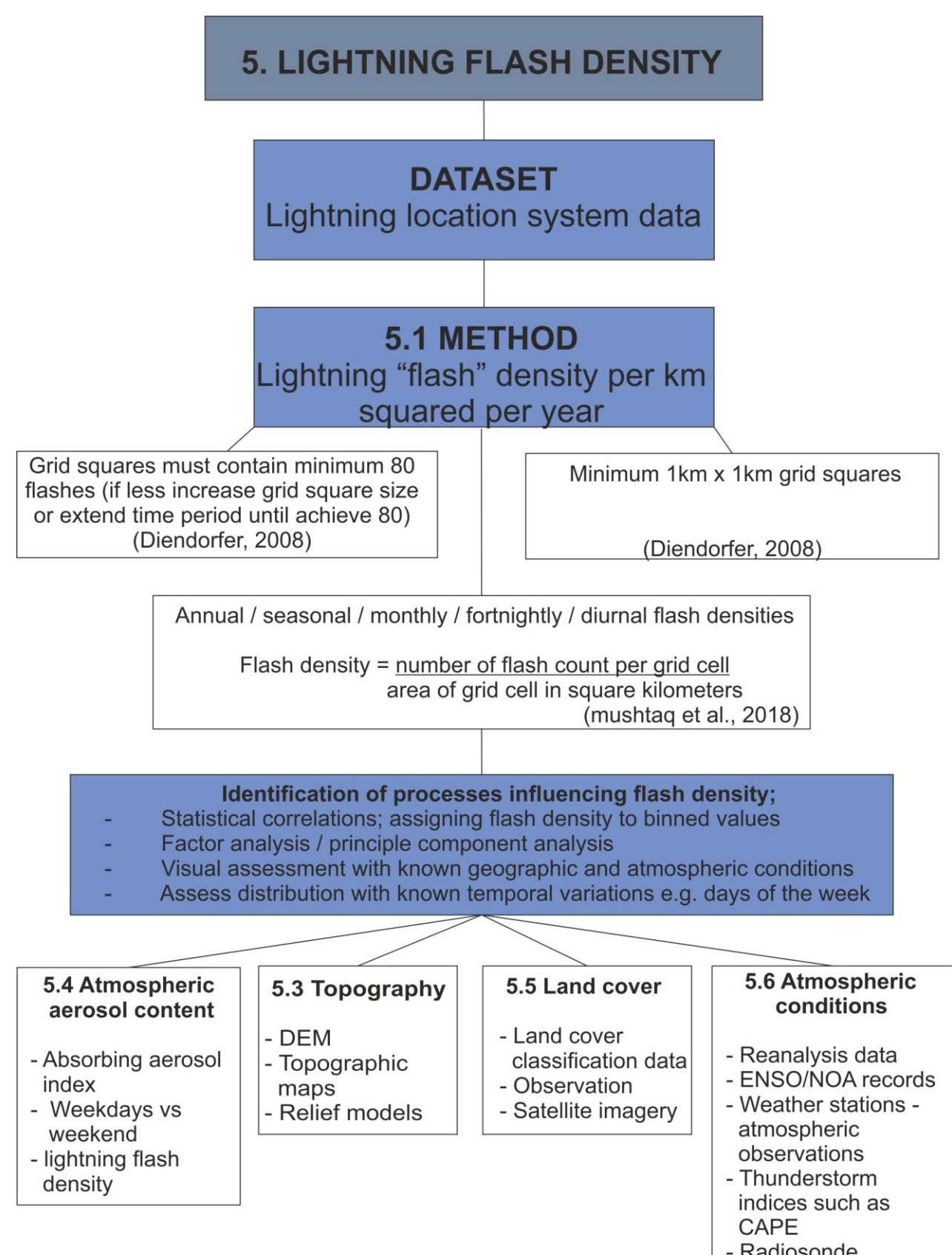

**Figure 4:  Diagrammatic summary of the steps involved in producing a lightning flash density thunderstorm climatology, showing the different variables to consider during the study design stage.**

List of Acronyms

CAPE                Convective Available Potential Energy

DE                  Detection Efficiency

ENSO                El Nino Southern Oscillation

ERA                 European Reanalysis Data

GIS                 Geographic Information Systems

hPa                 Hectopascal Pressure Unit

LA                  Location Accuracy

LI                  Lifted Index

MCS                 Mesoscale Convective Systems

NAO                 North Atlantic Oscillation

Synop               Surface Synoptic Observations

TORRO               The Tornado and Storm Research Organisation

TRMM LIS            Tropical Rainfall Measuring Mission Lightning Imaging Sensor

TRMM OTD            Tropical Rainfall Measuring Mission Optical Transient Detector

WWLLN               World Wide Lightning Location Network