# Peer review of "Review Article: A comprehensive review of datasets and methodologies employed to produce thunderstorm climatologies"

_Natural Hazards and Earth System Sciences, 2020_

## Referee Comment (RC1) · Tomeu Rigo (Referee) · 12 Mar 2020

General comments

The manuscript presents different a state-of-art of different ways of developing a systematic data base of episodes of thunderstorms, dividing the methodologies in three main groups: thunderstorm frequency, thunderstorm tracking or lightning flash density. The authors show the main advantages and disadvantages of each technique.

Although the idea of the work results interesting, there is a lack of coherence in the style

of the manuscript as a scientific publication. For instance, I can't find the motivation or the goal that leads to start the research. This is basic, because it helps to define the path that you choose along your research. What I mean is that if you want to present some type of methodologies, you need to clarify if the purpose of each one fits to your necessities, and which are the main disadvantages. And this is not appearing in any place of your manuscript.

Please, rewrite this condition and adapt all the rest of the manuscript to your necessities.

---

## Referee Comment (RC2) · Elissavet Galanaki (Referee) · 12 Apr 2020

The paper reviews the datasets and methodologies which have been used to produce thunderstorm climatologies. Precisely, it analyses the different implemented approaches for the computation of thunderstorm frequency, thunderstorm tracking or lightning flash density. The study concludes that the best choice of the applied method is related to the coverage and the quality of the available dataset and the controlling factors under investigation. The manuscript offers an additional contribution to understanding the advantages and disadvantages of the available methodologies applied in

thunderstorm climatology. However, the manuscript needs to be more organized in some sections. Thus, I recommend the minor revision of the manuscript before it is published. My detailed comments are given below:

1. The main objective and the motivation of this paper must be more clearly explained in the manuscript. 2. A paragraph at the end of the introduction that informs about the following structure of this manuscript must be added. 3. The "data section" must be in a separate section, not at the section of thunderstorm climatology. 4. Sometimes the authors write the lightning density as " flash density", sometimes as "lightning flash density" or " lightning density". One terminology must be selected.

Technical corrections

1. Title: without a full stop at the end. 2. Abstract: ". . . influenced by dataset coverage, quality and the controlling factors under investigation." What quality do you mean? Something is missing. . . 3. Table 1 2.14 The statement "Can be to produce longer climatologies for..." must be rephrased "Can be used to reconstruct activity in areas of poor coverage (Allen and Karoly, 2014)". Do you mean lightning activity? 4. You must explain the acronym CAPE at Line 103, not at Line 108. 5. Lines 163-165: You have already mentioned about radar reflectively in section 2.1.2 6. Line 186: "(Wapler and James, 2015) showed that 2 lightning strokes within a 15km radius was found to be the most effective.", without parenthesis at the names. 7. Line 327: Please, rephrase it.

---

## Author Comment (AC1) · 26 May 2020

Comment:

"There is a lack of coherence in the style of the manuscript as a scientific publication. For instance, I can't find the motivation or the goal that leads to the start of the research. This is basic, because it helps to define the path that you choose along your research. What I mean is that if you want to present some type of methodologies, you need to clarify if the purpose of each one fits your necessities, and which are the main

disadvantages. And this is not appearing in any place in your manuscript.

Please, rewrite this condition and adapt all the rest of the manuscript to your necessities."

Response:

To address the point raised we have restructured the paper and included two new paragraphs (below) at the end of the introduction providing the motivation for this review and making the justification clearer. We have also outlined the structure of the paper which illustrates how the subject areas covered address the goals of the paper. The two additional paragraphs read as follows:

"The purpose of the paper is to conduct a systematic and comprehensive review of the datasets and methodologies applied to create thunderstorm climatologies. This review aims to assist those at the design stage of their research and those new to the subject area to become familiar with the strengths and weaknesses of the available data types, to consider which climatological approach best fits their research goal and to identify potential alternative approaches which may not have previously been considered. Whilst there are existing reviews in this subject area available (Betz et al., 2009; Cummins and Murphy, 2009; Ellis and Miller, 2016; Nag et al., 2015), these tend to focus either on analysis of a particular dataset, data type or methodology. This paper, in contrast, fills a gap in the literature by providing an overview of the whole subject area to help the reader to subsequently move on to more specific and detailed examples. Lastly, recommendations for research areas which require development are made.

To fulfil the above purposes, we first review the dataset types in section 2, before then moving on to evaluating how different dataset types have been applied in compiling thunderstorm frequency climatologies (section 3) and thunderstorm tracking (section 4). Section 5 reviews the methods used to produce lightning flash density climatologies, using one dataset type: lightning remote sensing data. This section also includes a review on how lightning flash density results have correlated with potential drivers of

thunderstorm formation, such as topography, which thereby introduces further methods and datasets. Recommendations for study design are contained in Section 6 and future research areas outlined in Section 7."

---

## Author Comment (AC2) · 26 May 2020

Comment:

"The main objective and the motivation of this paper must be more clearly explained in the manuscript."

Response:

Added penultimate paragraph to introduction:

[Figure]

"The purpose of the paper is to conduct a systematic and comprehensive review of the datasets and methodologies applied to create thunderstorm climatologies. This review aims to assist those at the design stage of their research and those new to the subject area to become familiar with the strengths and weaknesses of the available data types, to consider which climatological approach best fits their research goal and to identify potential alternative approaches which may not have previously been considered. Whilst there are existing reviews in this subject area available (Betz et al., 2009; Cummins and Murphy, 2009; Ellis and Miller, 2016; Nag et al., 2015), these tend to focus either on analysis of a particular dataset, data type or methodology. This paper, in contrast, fills a gap in the literature by providing an overview of the whole subject area to help the reader to subsequently move on to more specific and detailed examples. Lastly, recommendations for research areas which require development are made."

Comment:

"A paragraph at the end of the introduction that informs about the following structure of this manuscript must be added."

Response:

Added final paragraph to introduction:

"To fulfil the above purposes, we first review the dataset types in section 2, before then moving on to evaluating how different dataset types have been applied in compiling thunderstorm frequency climatologies (section 3) and thunderstorm tracking (section 4). Section 5 reviews the methods used to produce lightning flash density climatologies, using one dataset type: lightning remote sensing data. This section also includes a review on how lightning flash density results have correlated with potential drivers of thunderstorm formation, such as topography, which thereby introduces further methods and datasets. Recommendations for study design are contained in Section 6 and future research areas outlined in Section 7."

Comment:

"The "data section" must be in a separate section, not at the section of thunderstorm climatology."

Response:

Amended section order: 1. Introduction 2. Data 2.1 Manual records 2.2 Thunderstorm remote sensing 2.3 Lightning remote sensing 2.4 Thunderstorm indices 3. Thunderstorm frequency 3.1 Manual records 3.2 Thunderstorm remote sensing 3.3 Lightning remote sensing 3.4 Thunderstorm indices 4. Thunderstorm frequency 4.1 Manual records 4.2 Thunderstorm remote sensing 4.3 Lightning remote sensing 5. Lightning flash density 5.1 Lightning flash density method 5.2 Global lightning flash density 5.3 Lightning flash density and topography 5.4 Lightning flash density and aerosols 5.5 Lightning flash density and land cover 5.6 Lightning flash density and atmospheric conditions 6. Recommendations 6.1 Dataset choice 6.2 The benefits of different types of approach 6.3 Identify the end user 7. Conclusion 7.1 Low-lightning areas 7.2 Dataset combination techniques 7.3 Reanalysis indices 7.4 Hazard communication and warnings

Comment:

"Sometimes the authors write the lightning density as "flash density", and sometimes as "lightning flash density" or "lightning density". One terminology must be selected."

Response: Amended, lightning flash density selected as single terminology.

Comment:

"Title: without a full stop at the end."

Response: Removed as requested.

Comment:

"Abstract: "..influenced by dataset coverage, quality and the controlling factors under investigation." What quality do you mean? Something is missing."

Response:

Amended as follows: "Regardless of approach, the choice of analysis method is heavily influenced by the coverage and quality (detection efficiency and location accuracy) of available datasets as well as by the controlling factors which are under investigation."

Comment:

"Table 1 2.14 The statement "Can be to produce longer climatologies for…" must be rephrased "Can be used to reconstruct activity in areas of poor coverage (Allen and Karoly, 2014)". Do you mean lightning activity?"

Response:

Amended as follows: "Can produce longer climatologies (Brooks et al., 2003)" "Can be used to reconstruct thunderstorm activity in areas of poor coverage (Allen and Karoly, 2014)"

Comment:

"You must explain the acronym CAPE at line 103, not at line 108."

Response:

Amended as follows: "Another approach is to calculate average daily values of relevant reanalysis variables such as 500 hPa and 1000 hPa geopotential heights, 500 hPa air temperature and the instability index known as CAPE (Convective Available Potential Energy) for a given temporal resolution (Gatidis et al., 2008). Reanalyses can also be used to obtain a longer climatology of thunderstorms by developing indices as proxies of thunderstorm activity (Kaltenböck et al., 2009; Kunz, 2007). Different indices may be more or less successful either in general or in different regions and seasons. An example of a commonly used index is CAPE which uses two of the three main ingredients for deep moist convection (namely instability and moisture) to evaluate the thunderstorm potential of environmental conditions (Moncreiff and Miller, 1976). The numerical CAPE value indicates the atmospheric potential to produce thunderstorms either looking at current conditions for forecasting or reconstructing the atmospheric conditions of the past for climatology (Holley et al., 2014)."

Comment:

"Lines 163-165: You have already mentioned about radar reflectivity in section 2.1.2"

Response:

Removed. Paragraph now begins: "Radar reflectivity value is used to provide data in relation to severity of convective events including thunderstorms and to diagnose mesoscale convective systems (Punkka and Bister, 2015); catalogue the percentage of thunderstorms that become intense; and identify thunderstorm initiation times and duration (Mohee and Miller, 2010)."

Comment:

"Line 186: "(Wapler and James, 2015) showed that 2 lightning strokes within a 15km radius was found to be the most effective.", without parenthesis at the names."

Response:

Amended as follows: "A successful threshold can be verified with alternative datasets such as human observation and radar; Wapler and James, (2015) showed that 2 lightning strokes within a 15km radius was found to be the most effective."

Comment:

"Line 327: Please, rephrase it."

Response:

Amended as follows: "Lightning flash density studies use data from lightning location

systems and some standardised analysis methods of best practice have been developed when using these datasets. Whilst most lightning climatologies are produced with the intention of minimising exposure to cloud-to-ground lightning hazards (Finke 1999), lightning climatologies can also be viewed as a form of thunderstorm climatology because lightning can be used to confirm thunderstorm activity."

―――――――――――――――――――

---

## Author Response (AR1)

| Reviewer | Reviewer comment | Author response |
|---|---|---|
| RC1 | There is a lack of coherence in the style of the manuscript as a scientific publication. For instance, I can't find the motivation or the goal that leads to the start of the research. This is basic, because it helps to define the path that you choose along your research. What I mean is that if you want to present some type of methodologies, you need to clarify if the purpose of each one fits your necessities, and which are the main disadvantages. And this is not appearing in any place in your manuscript.

Please, rewrite this condition and adapt all the rest of the manuscript to your necessities. | To address the point raised we have restructured the paper and included two new paragraphs (below) at the end of the introduction providing the motivation for this review and making the justification clearer. We have also outlined the structure of the paper which illustrates how the subject areas covered address the goals of the paper.

[revised manuscript text omitted]

**2.4**
**LIGHTNING FLASH DENSITY**

**DATASET**
**Lightning location system data**

Grid squares must contain minimum 80 flashes (if less increase grid square size or extend time period until achieve 80) (Diendorfer, 2008)

**2.4.1**
**METHOD**
**Lightning "flash" density per km² per year**

Minimum 1km x 1km grid squares (Diendorfer, 2008)

Annual / seasonal / monthly / fortnightly / diurnal flash densities

Flash density = Number of flash count per grid cell
Area of grid cell in square kilometers
(Mushtaq et al., 2018)

**Identification of processes influencing flash density;**
- Statistical correlations; assigning flash density to binned values
- Factor analysis / principle component analysis
- Visual assessment with known geographic and atmospheric conditions
- Assess distribution with known temporal variations e.g. days of the week

**2.4.3 Topography**

DEM
Topographic maps
Relief models

**2.4.4 Atmospheric aerosol content**

Absorbing aerosol index
Weekdays vs weekend lightning flash density

**2.4.5 Land cover**

Land cover classification data
Observation
Satellite imagery

**2.4.6 Atmospheric conditions**
Reanalysis data
ENSO / NOA records
Weather stations – atmospheric observations
Thunderstorm indices such as CAPE
Radiosonde

**Figure 4: Diagrammatic summary of the steps involved in producing a lightning flash density thunderstorm climatology and the different variables to consider during the study design stage.**

List of Acronyms

| | |
|---|---|
| CAPE | Convective Available Potential Energy |
| DE | Detection Efficiency |
| ENSO | El Nino Southern Oscillation |
| ERA | European Reanalysis Data |
| GIS | Geographic Information Systems |
| hPa | Hectopascal Pressure Unit |
| LA | Location Accuracy |
| LI | Lifted Index |
| MCS | Mesoscale Convective Systems |
| NAO | North Atlantic Oscillation |
| Synop | Surface Synoptic Observations |
| TORRO | The Tornado and Storm Research Organisation |
| TRMM LIS | Tropical Rainfall Measuring Mission Lightning Imaging Sensor |
| TRMM OTD | Tropical Rainfall Measuring Mission Optical Transient Detector |
| WWLLN | World Wide Lightning Location Network |

---

## Author Response (AR2)

**Author's response**

Reviewers comment:

> Dear authors,
>
> you have been done a good job improving your manuscript. I have just found that there is a limited number of references in some parts of the text. Here there are some suggestions:

Response:

> The majority of the suggested papers have now been included in text where highlighted below:

[revised manuscript text omitted]